



# Quantification of calcium carbonate (ikaite) in first– and multi–year sea ice

Heather Kyle[1,2], Søren Rysgaard[1, 2, 3,4], Feiyue Wang[2] and Mostafa Fayek[1]

[1]Department of Geological Sciences, University of Manitoba, Winnipeg, MB, R3T 2N2, Canada

[2]Centre for Earth Observation Science, University of Manitoba, Winnipeg, MB, R3T 2N2, Canada

[3]Arctic Research Centre, Aarhus University, 8000 Aarhus, Denmark

[4]Greenland Climate Research Centre, Greenland Institute of Natural Resources, 3900 Nuuk, Greenland

*Correspondence to:* Heather Kyle (umkyleh@myumanitoba.ca)



**Abstract**

Ikaite ($CaCO_3 \cdot 6H_2O$) is a metastable calcium carbonate mineral that forms at low temperature and/or high pressure. Ikaite precipitates in sea ice and may play a significant role in air–sea $CO_2$ exchange in ice covered seas and oceans. However, the spatial and temporal dynamics of ikaite in sea ice are poorly understood due to few available measurements and time consuming analytical techniques. Here, we present a new method for quantifying ikaite in sea ice and compare it with a more time-consuming imaging technique currently in use. In short, sea ice cores were

melted at low temperatures (<4°C), filtered for ikaite crystals that subsequently were dissolved and analyzed as dissolved inorganic carbon (DIC). The new method was applied on cores from experimental sea ice in Winnipeg (49° N), Canada, first–year sea ice near Cambridge Bay (69° N), Nunavut, Canada, and first– and multi–year sea ice near Station Nord (81° N), Greenland. Ikaite crystals were found in all sea ice types. The new ikaite quantification method is a straightforward technique that generally agrees with the image analysis technique and is both more

accurate and precise. The DIC method may give lower concentrations in first–year ice and higher concentrations in multi–year ice than image analysis, likely due to the large spatial variability of ikaite crystals in first–year sea ice and the small crystal size in multi–year ice, both of which make quantification by image analysis more difficult. The new method showed high concentrations of ikaite in 20 cm thick young sea ice (335 µmol kg$^{-1}$), lower concentrations in 1.5 m thick first–year sea ice (45 µmol kg$^{-1}$) and low concentrations in 3.3 m thick multi–year sea

ice (3 µmol kg$^{-1}$). Highest concentrations were observed in the upper ice layers at all stations and layers where sea ice algae were present. The higher abundance of ikaite in young first–year sea ice indicates that its concentrations will likely increase in the Arctic as a result of the recent rapid decline of the multi–year ice cover and increasing presence of seasonal sea ice. As a result, it is likely that ikaite will play a more significant role in air–sea $CO_2$ exchange in ice–covered seas in the future.

**1 Introduction**

Sea ice cover in the Arctic has often been considered to act as a barrier to air–sea $CO_2$ exchange (Tison et al., 2002). However, recent studies (e.g., Rysgaard et al., 2007; 2011; Geilfus et al., 2013a) have indicated that this is not the case and that processes driving air–sea gas exchange are active throughout the sea ice season. During sea ice formation, dissolved inorganic carbon (DIC) is released from sea ice to the underlying water, decreasing DIC and total alkalinity (TA) concentrations in sea ice (Rysgaard et al., 2007). During the winter months, this dense brine

sinks to deeper waters, lowering the DIC, TA, and $CO_2$ concentrations of the surface water. Ice melt will further lower $CO_2$ concentrations in the surface ocean, increasing air to ocean $CO_2$ fluxes. This process is referred to as the sea ice carbon pump (Rysgaard et al., 2011). Recent studies (e.g., Grimm et al., 2016) have indicated that the sea ice carbon pump may be significant on a regional scale, where it is largely determined by the cycle of sea ice growth

and decay. In addition, freezing concentration can result in the formation of carbonate precipitates in sea ice brine (Geilfus et al., 2013a; Papadimitriou et al., 2014). Ikaite ($CaCO_3 \cdot 6H_2O$), a metastable calcium carbonate mineral,



precipitates out of sea ice at approximately –2.2ºC (Assur, 1958) under standard seawater conditions according to Eq. (1) (Rysgaard et al., 2013):

$$Ca^{2+} + 2HCO_3^- + 5H_2O = CaCO_3 \cdot 6H_2O + CO_2 \tag{1}$$

Brine drainage occurs during sea ice formation and melt, resulting in the removal of dissolved $CO_2$ and salts (Rysgaard et al., 2007; 2009; 2013; Parmentier et al., 2013). As long as the sea ice remains permeable, brine drainage continues, and any $CO_2$ generated during ikaite precipitation escapes the sea ice system (Rysgaard et al., 2009). Ikaite crystals remain trapped in the ice matrix throughout the winter and are released during sea ice melt (Rysgaard et al., 2007). During the winter, $CO_2$ is separated from ikaite crystals by diffusion, yielding a higher $CO_2$

escape to the underlying water relative to ikaite (Rysgaard et al., 2007). Ikaite has been observed in both Antarctic and Arctic sea ice (Dieckmann et al., 2008; 2010) and may play a significant role in the sea ice carbon pump (Rysgaard et al., 2011; Parmentier et al., 2013), but the spatial and temporal dynamics of ikaite in sea ice are poorly understood (Rysgaard et al., 2014). Ikaite that remains trapped in the sea ice matrix during the ice growth season will store TA in the sea ice, which increases the buffering capacity of sea ice and meltwater and becoming a source

of excess TA to seawater when ikaite dissolves during sea ice melt, enhancing $CO_2$ uptake by the ocean (Rysgaard et al., 2007; 2009). In addition, the dissolution of ikaite will consume $CO_2$, lowering the surface water $pCO_2$ and further enhancing the $CO_2$ flux from the atmosphere to the ocean during sea ice melt (Rysgaard et al., 2013).

Arctic summer sea ice extent and volume is rapidly decreasing, with seasonal sea ice cover replacing multi–year sea ice (Stroeve et al., 2014). Nine of the lowest sea ice extents on record occurred between 2007 and 2015 (Galley et

al., 2016) and it is predicted that the Arctic could be ice free in September before 2050 (Overland and Wang, 2013). The increased first–year sea ice cover could greatly increase the significance of the sea ice carbon pump when considering carbon fluxes in ice covered seas (Grimm et al., 2016). This is particularly true when considering carbon fluxes on a regional scale, since the sea ice carbon pump is largely influenced by the cycle of sea ice growth and decay, as well as changes in sea ice porosity, temperature, and salinity (Rysgaard et al., 2011). Since ikaite most

commonly precipitates in first–year sea ice, increasing seasonal sea ice cover will increase its contributions to air–sea $CO_2$ exchange in ice covered seas.

To improve our understanding of its role in the sea ice carbon pump, a simple and reliable method of quantifying ikaite must be developed. The method currently in use consists of image analysis of small quantities of ice followed by ikaite confirmation by x–ray diffraction analysis (Rysgaard et al., 2013). This is a very time consuming process

that requires sophisticated equipment and can potentially underestimate ikaite concentration when the crystals are too small. Here we present a new, faster method for ikaite analysis based on measurements of DIC in filtered ice samples. The method was applied to various types of experimental and natural sea ice and compared with the image analysis technique. This method has the potential to rapidly expand the database of ikaite measurements in sea ice, which will greatly improve our understanding of regional and global importance of the sea ice carbon pump. In

addition, the vertical distribution as well as the amount of ikaite in different sea ice types is discussed based on other

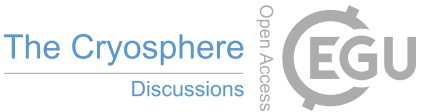

measured environmental parameters to determine how changes in sea ice cover in the Arctic may affect ikaite concentration.

## 2 Methods

### 2.1 Study sites and sampling

Sea ice cores were collected from three sites: the Sea–ice Environmental Research Facility (SERF), Winnipeg, Canada in January 2013, Station Nord, northeastern Greenland in April 2015, and Cambridge Bay, Nunavut, Canada in May 2016 (Table 1). The pool at SERF acts as a sea ice–seawater mesocosm; experimental sea ice is grown from artificial seawater in an 18.3 m long, 9.1 m wide, and 2.6 m deep in–ground outdoor concrete pool exposed to ambient temperatures, winds, and solar radiation (Geilfus et al., 2016). To mimic the major composition of natural

seawater, the pool is filled with large quantities of various rock salts dissolved into local groundwater (Rysgaard et al., 2014). At Station Nord, first–year sea ice grows from seawater with a large freshwater content, with relatively thin (<1 m) near shore and thicker multi–year sea ice further offshore (Bendtsen et al., 2017). Sea ice at Station Nord was overlain by a thick snow cover (>1 m) during data collection. Sea ice at Cambridge Bay forms from seawater with a mean salinity of ~30 and melts annually. Sea ice is typically ~1.6 m thick and is overlain by a variable snow

cover, typically ranging from 5–30 cm thick.

Samples were obtained from one site at SERF, four first and two multi–year sites at Station Nord, and six first–year sites at Cambridge Bay. At each site, two sea ice cores were extracted using a Mark II 9 cm coring system (Kovacs Enterprises). At each sampling site, vertical temperature profiles were obtained using a calibrated temperature probe on one of the cores collected for analysis. Each sea ice core was cut into 10 cm sections. In the laboratory, each

section from one core was weighed and stored in a sealed plastic bag in a –20ºC freezer until they could be processed (usually within 24 hours). The sections from the second sea ice core were weighed, placed in vacuum–sealed plastic bags, and allowed to melt at 2ºC immediately upon arriving at the laboratory from the field.

### 2.2 Analysis

The new DIC method of ikaite quantification was compared with the image analysis technique developed by

Rysgaard et al. (2013). One sea ice core from each sampling site was used to test each technique.

#### 2.2.1 Image analysis

The details of the image analysis technique can be found in Rysgaard et al. (2013). In brief, for each 10 cm section of sea ice core, three 46 to 567 mg subsamples of sea ice were cut off using a stainless steel knife at random places and weighed. Each subsample was placed on a chilled glass slide resting on a chilled aluminium block with a 1 cm

viewing hole. In a 20ºC laboratory, subsamples were examined under a Leica DMiL LED microscope at 100x magnification as they were allowed to melt. A few seconds after the ice completely melted, three photographs were taken at 100x magnification of random areas on the slide using a Leica DFC 295 camera and Leica Application





Suite version 4.0.0 software. Three to five minutes later, a second set of photographs were taken in the same locations; any crystals that had begun to dissolve were assumed to be ikaite. Samples were analyzed by x–ray

diffraction to confirm the presence of ikaite in the sea ice samples (Rysgaard et al., 2013).

Using the first set of photographs, the abundance and concentrations of ikaite crystals were calculated from the images using the software ImageJ (version 1.49g). Each image was brightness/contrast adjusted and converted to binary. "Close" and "fill holes" functions were applied to each image so all ikaite crystals were filled with black. An "analyze particles" function was applied to count the crystals and to determine the area of each photograph covered

by ikaite. The area was then converted to volume assuming a cubic mineral structure for ikaite. Using the calculated volume, density (1.78 g cm$^{-3}$) and molar mass (208.18 g mol$^{-1}$) of ikaite (Rysgaard et al., 2013), and the mass of the subsample, ikaite concentrations were calculated and converted to μmol kg$^{-1}$ melted sea ice, according to Eq. (2):

$$[\text{ikaite}] = \frac{\left(1.0 \times 10^6 \left(\frac{\rho V_{\text{ikaite}}}{M}\right)\right)}{\left(\frac{m_{\text{ice}}}{1000}\right)} \tag{2}$$

where $\rho$ is the density in g cm$^{-3}$, M is the molar mass in g mol$^{-1}$, $V_{\text{ikaite}}$ is the volume of ikaite crystals in cm$^3$ and $m_{\text{ice}}$ is the mass of the subsample in g. The numerator is multiplied by $1.0 \times 10^6$ to convert mol to μmol and the

denominator is divided by 1000 to convert g to kg. For each 10 cm section, the average concentration of each subsample was determined to estimate the overall concentration of each section. After image analysis was complete, the sea ice core was melted at 2ºC to ensure ikaite crystals did not dissolve. As soon as melting was complete, 50 ml were withdrawn and warmed to 20ºC to determine bulk salinity using an Orion 3–star with an Orion 013610MD conductivity cell. The remaining meltwater was used for duplicate samples for the DIC method of ikaite

quantification (Sect. 2.2.2).

### 2.2.2 DIC and TA analysis

To prepare the sea ice cores for the DIC method of ikaite quantification, each 10 cm section was completely melted in a sealed plastic bag at 2ºC to ensure ikaite crystals did not dissolve. As soon as melting was complete, four 12 ml Exetainers (Labco Limited, High Wycombe, UK) were filled with meltwater for DIC and total alkalinity (TA)

analysis. The remaining meltwater was filtered through a Whatman GF/F 47 mm glass microfiber filter; the filter containing ikaite crystals was placed in a 12 ml Exetainer containing a glass bead that was then filled with deionised water. Additional Exetainers were filled with deionised water to use as blanks. All Exetainers were poisoned with 12 μl saturated HgCl$_2$ solution and shipped to the Centre for Earth Observation Science (CEOS) at the University of Manitoba for further analysis.

The TA and DIC analyses were completed at the University of Manitoba. Dissolved inorganic carbon analysis was done using an Apollo SciTech DIC analyzer with a solid state infra–red CO$_2$ detector (LI 7000) and TA analysis was completed using a TIM 840 potentiometric titrator. Certified reference materials supplied by the Scripps Institution of Oceanography were routinely entered into the sample stream to ensure accuracy of results.



Prior to analyzing the Exetainers containing the filtered crystals for DIC concentrations, 100 μl of 1 M HCl was added to ensure that all carbonate crystals were completely dissolved. The average DIC concentration of the blanks was subtracted from each value from the DIC of the filtered crystals. Ikaite concentrations were calculated according to Eq. (3):

$$[ikaite] = \frac{DIC_{filtered}\, m_{ex}}{\dfrac{m_{ice}}{1000}} \tag{3}$$

where $DIC_{filtered}$ is the DIC of the filtered ikaite crystals in μmol kg$^{-1}$, $m_{ex}$ is the mass of deionised water added to the Exetainer in g, and $m_{ice}$ is the mass of filtered meltwater in g. Dissolved inorganic carbon can be used as a proxy for ikaite when using this method since the dissolution of ikaite crystals at temperatures greater than 4ºC produces $HCO_3^-$, a major component of DIC (Geilfus et al., 2012).

## 3 Results

### 3.1 Ikaite

Image analysis showed that the euhedral ikaite crystals ranged from 5 to ~100 μm on the longest side (Fig. 1). The majority of crystals had a distinct rhombic structure; crystals smaller than 10 μm began to dissolve quickly and had a more rounded appearance. In first–year sea ice, ikaite was most abundant in the upper 20 cm of the ice, with lower concentrations in the middle ice layers and a small increase in ikaite abundance in the lower 20 cm. This trend was not observed in multi–year sea ice, which had relatively constant low ikaite concentrations throughout. Ikaite and ice algae were present together in the lower 20 cm of sea ice cores from Cambridge Bay.

Ikaite concentrations were calculated at SERF using image analysis only at 1 cm intervals (Fig. 2). Concentrations ranged from 3.9±3.3 to 717.3±138.3 μmol kg$^{-1}$ (unless otherwise specified, all concentrations are given in μmol kg$^{-1}$ melted sea ice and errors represent the standard error of the mean throughout). Sea ice at SERF was thinner than that at both Station Nord and Cambridge Bay and ikaite concentrations at SERF were higher than at the other sampling sites.

Ikaite concentrations were determined using image analysis and DIC analysis of filtered crystals at all other sampling sites. Mean ikaite concentrations are shown in Figures 3–5, although it should be noted that all calculations concerning the agreement of the two methods were completed using the ikaite concentrations of individual sea ice cores. In Figures 4 and 5, the two methods do not always appear to agree, although the difference is small when this is the case (i.e., less than 8 μmol kg$^{-1}$ in Figure 4 and less than 20 μmol kg$^{-1}$ in Figure 5). Concentrations in first–year sea ice determined with the DIC method ranged from 2.5±0.0 to 11.9±5.7 μmol kg$^{-1}$ and those at Cambridge Bay ranged from 32.6±6.0 to 69.5±30.3 μmol kg$^{-1}$. Using image analysis at the same sampling sites, mean ikaite concentrations ranged from 0.7±0.0 to 14.3±8.6 μmol kg$^{-1}$ in first–year sea ice at Station Nord and 5.5±2.9 to 261.7±153.3 μmol kg$^{-1}$ in first–year sea ice at Cambridge Bay. In multi–year sea ice at Station Nord, the DIC method generally yielded slightly higher concentrations than image analysis, and the DIC and image analysis



methods yielded mean concentrations ranging from 1.9±0.4 to 7.8±0.0 µmol kg$^{-1}$ and from 0.0±0.0 to 1.4±1.0 µmol kg$^{-1}$, respectively, in the upper 170 cm of the sea ice. Average ikaite concentrations calculated with both methods at each sampling site are shown in Table 2.

At both Station Nord and Cambridge Bay, a small (4–12 m$^2$) area was cleared of snow and first–year sea ice cores were collected from these areas two days later to test how ikaite concentrations would be affected. At Station Nord,

only the upper 39 cm of the core from the snow cleared area was examined and ikaite concentrations ranged from 8.7±6.3 to 47.7±44.3 µmol kg$^{-1}$, which is a slight increase in concentration from cores collected from the same area when snow was not removed. Full sea ice cores were extracted from the snow cleared area at Cambridge Bay; ikaite concentrations in this area ranged from 0 to 30.3±13.1 µmol kg$^{-1}$, which is not significantly different than concentrations from the same area where snow cover remained in place.

**3.2 Environmental data**

Air temperature and snow and ice thickness were recorded at all sampling sites. Air temperature ranged from −30.7 to −17.2 during the SERF experiment in January 2013, from −27.2 to −10.2℃ during the Station Nord field campaign in April 2015, and from −6.4 to −0.6℃ during the Cambridge Bay field campaign in May 2016. Snow was removed from the sea ice pool during the 2013 SERF experiment; snow thicknesses ranged from 79 to 135 cm at

Station Nord in April 2015 and from 2 to 30 cm at Cambridge Bay in May 2016. Average sea ice thickness ranged from ~20 cm at SERF to 174 cm at Cambridge Bay (Table 2). Due to the capability of ice coring equipment, it was not possible to collect full cores when sea ice thickness exceeded 200 cm.

Sea ice temperature, bulk salinity, TA, and DIC vertical profiles were measured in each sea ice type (Figs. 6–9). Sea ice temperatures ranged from −13.0 to −3.9℃ at SERF, from −5.0 to −0.5℃ in first–year sea ice at Station Nord,

from −7.2 to −0.7℃ in multi–year sea ice at Station Nord, and from −7.3 to −1.1℃ in first–year sea ice at Cambridge Bay. In the snow cleared areas at Station Nord and Cambridge Bay, sea ice temperatures ranged from −9.2 to −5.2℃ and from −3.8 to −1.3℃, respectively. Warmer ice temperatures, such as those at Station Nord, were associated with thicker snow cover. Bulk salinities ranged from 4.9 to 24.0 at SERF, from 0.5 to 6.5 in first–year sea ice at Station Nord, from 0.0 to 0.5 in multi–year sea ice at Station Nord, and from 3.1 to 9.3 at Cambridge Bay. In the snow

cleared areas, bulk salinity ranged from 1.9 to 3.9 at Station Nord and from 3.6 to 6.2 at Cambridge Bay. Mean temperature and bulk salinity profiles are shown in Figs. 6a, 7a, 8a, and 9a.

At SERF, TA ranged from 376.2 to 1433.2 µmol kg$^{-1}$ and DIC ranged from 316.9 to 862.0 µmol kg$^{-1}$, with TA:DIC ratios ranging from 1.1 to 1.7. In thicker sea ice, TA and DIC concentrations were more variable than at SERF (Figs. 7b, 8b, 9b).In general, the TA:DIC ratio increased toward the ice–water interface. At Station Nord, TA ranged from

25.1 to 580.1 µmol kg$^{-1}$ in first–year sea ice and from 7.0 to 183.0 µmol kg$^{-1}$ in multi–year sea ice; DIC ranged from 82.0 to 510.9 µmol kg$^{-1}$ in first–year sea ice and from 27.0 to 236.2 µmol kg$^{-1}$ in multi–year sea ice. It should be noted, however, that due to technical difficulties with recovering the bottom of sea ice cores longer than 200 cm, our measurements may be underestimated for thick multi–year sea ice. At Cambridge Bay, TA ranged from 232.7 to





552.4 µmol kg$^{-1}$ and DIC ranged from 168.5 to 555.3 µmol kg$^{-1}$. In the snow cleared area at Station Nord, TA

ranged from 410.2 to 580.1 µmol kg$^{-1}$, and DIC ranged from 367.5 to 510.9 µmol kg$^{-1}$. At Cambridge Bay, the snow

cleared area yielded TA and DIC concentrations ranging from 354.2 to 510.8 µmol kg$^{-1}$ and from 168.5 to 487.3

µmol kg$^{-1}$, respectively.

Sea ice algae were present in all ice cores at Cambridge Bay (mean chl-a concentration 119.2±26.1 µg l$^{-1}$; Figs. 1,

10) and occurred in the lower 5–15 cm of the sea ice cores. Sea ice algae were not observed at Station Nord and

were absent from SERF since biological activity was not deliberately introduced to the SERF pool.

### 4 Discussion

#### 4.1 Effectiveness of DIC analysis of filtered crystals to quantify ikaite

Errors in ikaite concentrations were calculated using the standard error of the mean. Based on mean ikaite

concentrations, when using image analysis, the average standard error was 19.4 µmol kg$^{-1}$; standard errors had a

large variation, which was likely due to the uneven distribution of ikaite crystals and the small subsample size. The

average standard error when using DIC analysis of filtered crystals was 5.1 µmol kg$^{-1}$, indicating that this method is

more precise than image analysis. The smaller errors when using the DIC method are because the full sea ice core is

used, so all ikaite crystals are included during analysis. When not considering outliers from image analysis

(concentrations of 0 µmol kg$^{-1}$ or greater than 400 µmol kg$^{-1}$), the two methods generally agreed within error (using

a paired t–test, p<0.05), indicating that DIC analysis of filtered crystals is an effective method of ikaite

quantification. The smaller errors from the DIC method indicate that it is a more reliable technique than image

analysis for determining ikaite concentrations in sea ice.

Although the techniques agree within error, the raw ikaite concentrations determined using the DIC method yielded

higher values than image analysis in all multi–year sea ice and in 47% of first–year sea ice samples. In multi–year

sea ice, this is likely due to the low number of observed ikaite crystals and the fact that the few crystals present were

quite small (i.e., <10 µm). Dieckmann et al. (2008) indicate that since ikaite is not stable at temperatures above 4ºC,

crystals will begin to dissolve immediately. As a result, any crystals that were smaller than ~5 µm may have

dissolved before they could be photographed. In first–year sea ice, ikaite crystals are larger (5 to 100 µm) and may

take several hours to dissolve completely (Rysgaard et al., 2012). This is consistent with observations made at

Cambridge Bay, where a 50 µm crystal dissolved completely in ~15 min. and a 75 µm crystal was greater than half

its original size after 20 minutes at room temperature. Crystals that completely or partially dissolve before they can

be photographed will result in an underestimation of ikaite concentration when using image analysis. However, it is

assumed that since meltwater never warmed to more than 4ºC during filtration, ikaite crystals remain intact, meaning

they will be included in calculations when using DIC analysis of filtered crystals to quantify ikaite. To prevent rapid

dissolution of crystals during image analysis, slides were placed on a chilled aluminium slide to keep meltwater

cold.





Image analysis is a time consuming process that requires additional equipment in the field (e.g., microscope) and involves extensive sample preparation and processing. Three subsamples are analyzed for each 10 cm sea ice section and each subsample takes approximately 10 minutes to examine since the ice needs to melt before it can be photographed. Each subsample then needs to be photographed and each photograph needs to be processed (software ImageJ) before ikaite concentration can be calculated. However, the full core is used when using DIC analysis of filtered crystals and very little sample preparation is required. Sea ice needs to be melted anyway for TA and DIC measurements and by ensuring that temperatures remain between 0ºC and 4ºC during sea ice melt and filtration, it can also be used for DIC analysis to determine ikaite concentration. Filtration takes about 2 minutes per sample so it is also less time consuming than image analysis.

Another limitation of image analysis is that ikaite is unevenly distributed through sea ice. Previous studies have indicated that ikaite crystals are mainly located in the interstices between ice platelets (Rysgaard et al., 2014), accounting for the uneven spatial distribution. Subsamples for image analysis are taken from random places throughout each 10 cm sea ice section to attempt to get a representative sample. However, each subsample weighs less than 600 mg, representing less than 1% of the total sample, so random subsamples may not be truly representative of the total ikaite concentration in sea ice. Furthermore, ikaite is unevenly distributed on a microscopic scale and ikaite is absent from the majority of each subsample. During microscopy, areas chosen to photograph tend to be those containing ikaite crystals. This could result in an overestimation of ikaite concentration when using image analysis. This bias is eliminated when using the DIC analysis method since the entire sea ice core is used, yielding a more accurate ikaite concentration.

The DIC analysis of filtered ikaite crystals is an effective method of ikaite quantification, but it still has some limitations. Crystals smaller than 0.7 μm will pass through the filter pores. However, ikaite crystals of this size were not detected by microscopic analysis and therefore not included in ikaite concentrations by either method. If crystals dissolve before filtration is completed, they will not be considered in calculations using DIC analysis of filtered crystals. However, by maintaining low temperatures (2ºC) during sea ice melt and by working quickly to filter the meltwater, dissolution of these crystals should be minimized. It is assumed that crystals will not dissolve below 4ºC, but in principle, dissolution may also occur if ikaite is in contact with atmospheric $CO_2$ as the ice melts (Rysgaard et al., 2012). Furthermore, DIC analysis includes all DIC in the sample, including any that is added to the sample when deionised water is added during sample preparation. To account for any DIC that is added to the sample, blanks are taken and the DIC concentration of the blanks are subtracted from the total DIC concentration before calculating ikaite concentration.

### 4.2 Relationship between ikaite and environmental data

Ikaite concentrations in sea ice are expected to increase with decreasing temperature (Rysgaard et al., 2013). During this study, the highest ikaite concentrations were generally observed in the upper 20 cm of the sea ice cores where the ice was coldest, which is consistent with ikaite concentrations elsewhere in the Arctic (e.g., Rysgaard et al., 2013; 2014).. In addition, the coldest sea ice, located at SERF, contained higher ikaite concentrations than other



sampling site in this study and previous observations. Sea ice temperatures recorded at Station Nord were higher than those at Cambridge Bay and SERF due to the thick insulating snow cover at Station Nord. These warm temperatures correspond with lower ikaite concentrations than those at Cambridge Bay and at SERF, as well as

those previously recorded such as 15 to 25 µmol kg$^{-1}$ near Barrow, Alaska (Geilfus et al., 2013a), 100 to 900 µmol kg$^{-1}$ in Young Sound (Rysgaard et al., 2013), and 162.1 to 241.5 µmol kg$^{-1}$ northeast of Greenland (Rysgaard et al., 2012). Nomura et al. (2013) did not observe a clear link between sea ice temperature and the presence or absence of ikaite crystals, but this study shows that warmer sea ice is less likely to contain high concentrations of ikaite. Colder sea ice temperatures were recorded in multi–year sea ice than in first–year sea ice at

Station Nord, but ikaite concentrations in multi–year sea ice were lower, indicating a lack of correlation between temperature and ikaite concentration in multi–year ice at Station Nord. The multi–year sea ice in Station Nord is formed from low salinity water originating from melting snow that retains summer melt below the uneven subsurface of the ice, so the brine salinity may not be high enough to facilitate ikaite production, which may not be the case for other multi–year ice forms further offshore. Lower ikaite concentrations in multi–year sea ice compared

with first–year sea ice at Station Nord are consistent with observations at previous studies (e.g., Rysgaard et al., 2009).

Changes in sea ice temperature can also impact ikaite concentrations on short time scales. In the snow cleared area at Station Nord, ice temperature dropped to –9.2°C from approximately –4.2°C at the upper ice interface. This corresponded with an increased ikaite concentration from 4.9±1.0 µmol kg$^{-1}$ to a mean of 27.9±8.2 µmol kg$^{-1}$ in the

upper 30 cm of the sea ice cores, indicating that the rapid cooling of the ice promoted ikaite precipitation. Due to warmer air temperatures at Cambridge Bay, the temperature of the upper sea ice surface warmed slightly (from approximately –5.2°C to –2.4°C) and the mean ikaite concentration remained constant within error with values of 45.3±3.4 µmol kg$^{-1}$ when snow cover remained in place and 40.7±5.2 µmol kg$^{-1}$ when the snow was removed. The rapid cooling of the sea ice at Station Nord when the snow cover was removed was due to the large difference

between the air and ice temperatures before snow removal (~10°C different). Ikaite concentrations at Cambridge Bay were not significantly affected by removing snow since temperatures did not change significantly as a result of the thinner snow cover. It should also be noted that since ikaite precipitates at short time scales when temperatures decrease, the storage of sea ice at –20°C prior to image analysis, may produce artifact ikaite and influence the results. To minimize this effect, cores are kept at in situ temperature or stored as little as possible before processing.

Higher bulk salinity is expected to yield higher ikaite concentrations (Rysgaard et al., 2014). Typical C–shaped bulk salinity profiles were observed in all first–year sea ice; this was particularly pronounced in sea ice grown at SERF since the presence of brine skims and frost flowers near the air–ice interface made the upper ice surface bulk salinity much higher than the middle ice layers. In first–year sea ice, the C–shaped bulk salinity profiles also corresponded with  C–shaped ikaite concentration profiles (Figs. 2, 3, 5, 6, 7, 9), which were more pronounced in thin

experimental sea ice at SERF than in thicker first–year sea ice. The highest bulk salinities (i.e., at SERF) corresponded with the highest ikaite concentrations and the lowest bulk salinities (i.e., in multi–year sea ice at Station Nord) were associated with the lowest ikaite concentrations, indicating the two are closely related. Bulk



salinities in multi–year sea ice at Station Nord were near zero and ikaite concentrations were below 7 µmol kg$^{-1}$, indicating that in multi–year sea ice salinity influences ikaite precipitation more strongly than temperature.

It was previously suggested that bulk TA:DIC ratios of 2 would indicate ikaite precipitation (Rysgaard et al., 2007; 2009; Geilfus et al., 2016). However, ikaite crystals were observed in all ice types during this study and mean TA:DIC ratios ranged from 0.7 in multi–year sea ice at Station Nord to 1.3 at SERF (Figs. 6–9), indicating that ikaite precipitation will occur when the TA:DIC ratio is less than 2. This also supports previous findings, where ikaite was observed when TA:DIC ratios ranged from 1.1 to 1.6 (Geilfus et al., 2013a). In general, higher ikaite

concentrations are associated with higher TA:DIC ratios. The highest observed mean ikaite concentrations (717.3 µmol kg$^{-1}$ at SERF) were associated with the highest TA:DIC (1.6). At Cambridge Bay, the highest mean TA:DIC ratio (1.3) also corresponded with the highest ikaite concentration (69.5 µmol kg$^{-1}$).

Both TA and DIC concentrations will also affect ikaite concentration. The highest TA and DIC concentrations (1433.2 µmol kg$^{-1}$ and 862.0 µmol kg$^{-1}$, respectively at SERF) were also associated with the highest ikaite

concentrations. In thicker first–year sea ice at Cambridge Bay, TA, DIC, and ikaite concentrations showed general C–shaped vertical profiles (Figs. 5, 9). Ikaite concentration and TA are better correlated than ikaite concentration and DIC, indicating that much of the TA in the sea ice is stored in ikaite, as suggested by previous studies (e.g., Rysgaard et al., 2007; 2009).

At Station Nord, the mean TA:DIC ratio is less than one in both first– and multi–year sea ice (0.7 and 0.9,

respectively) and generally increases with depth, but ikaite concentration generally decreases with depth and were lower than in first–year sea ice at Cambridge Bay, SERF, and in previous studies (e.g., Geilfus et al., 2013a; Rysgaard et al., 2013). The low TA and DIC concentrations observed at Station Nord, combined with the low bulk salinity, are not conducive to ikaite precipitation, hence the low ikaite concentrations observed in both first and multi–year sea ice. Small variations in low TA and DIC concentrations (e.g., less than 100 µmol kg$^{-1}$ in multi–year

sea ice and less than 220 µmol kg$^{-1}$ in first–year sea ice) will cause large variations in the TA:DIC ratio, so ikaite concentration and TA:DIC ratios are not closely related when TA and DIC concentrations are low. The low TA:DIC ratios at Station Nord indicate that most TA is stored in the sea ice as ikaite but there is still $CO_2$ and DIC present. In interior sea ice layers, which act as a semi–closed system, ikaite precipitation produces $CO_2$ that could remain trapped, increasing DIC and $CO_2$ concentrations and lowering the pH and promoting ikaite dissolution (Hare et al.,

2013). This process may partially explain the low ikaite concentrations in first–year sea ice at Station Nord. However, low TA and DIC concentrations, in conjunction with low bulk salinity, will prevent supersaturation of sea ice brine with respect to carbonate and limit ikaite precipitation.

It is also possible that DIC concentrations are higher than TA concentrations due to heterotrophic bacterial activity in the sea ice, which would release $CO_2$ and increase DIC concentrations. More work is required to determine the

relationship between bacterial activity and ikaite precipitation, since available data is currently limited (e.g., Søgaard et al., 2013). It is also likely that during the sea ice algae bloom, DIC concentrations will decrease, resulting in an increased TA:DIC ratio (Rysgaard et al., 2007), which is consistent with observations at Cambridge Bay, where



higher chlorophyll–*a* concentrations were associated with low DIC concentrations and high TA:DIC ratios (Fig. 10). The increased TA:DIC ratio near the bottom of this sea ice core, where sea ice algae is present, is also

associated with an increased ikaite concentration (Figs. 1, 10). The elevated ikaite concentrations when sea ice algae are present may be because carbon is consumed during photosynthesis, increasing pH and enhancing the conditions that promote ikaite precipitation.

### 4.3 Influence of ikaite in ice covered seas

First–year sea ice typically has higher bulk salinity than multi–year sea ice, which is consistent with observations at

Station Nord, where both first and multi–year sea ice were sampled. Higher bulk salinity is also associated with higher TA and DIC concentrations (Sect. 4.2) and ikaite concentrations increase with increasing bulk salinity and TA and DIC concentrations. If ikaite crystals remain trapped in the sea ice but the $CO_2$ produced by ikaite precipitation is removed from the sea ice, either by gravity drainage (e.g., Rysgaard et al., 2007) or by being released to the atmosphere (e.g., Geilfus et al., 2013a), then the dissolution of ikaite during sea ice melt will result in the

release of excess TA that is stored in ikaite, increasing the buffering capacity of sea ice and surface meltwater and enhancing $CO_2$ uptake by the ocean (i.e., the sea ice driven carbon pump; Rysgaard et al., 2007; 2009; 2011). In the mid–1980s, multi–year sea ice accounted for 70% of Arctic winter sea ice extent but this dropped to less than 20% by 2012 (Stroeve et al., 2014). The replacement of multi–year sea ice with seasonal sea ice will result in increased salinity and TA and DIC concentrations, so ikaite concentrations are also expected to increase with increasing first–

year sea ice cover. Higher ikaite concentration will store higher amounts of TA, so it is likely that more first–year sea ice cover will enhance the sea ice carbon pump and increase the contribution of ice covered seas to global carbon fluxes (Grimm et al., 2016).

The exact timing of ikaite precipitation throughout the sea ice season is not well known, but changing sea ice conditions (e.g., decreased temperature or increased salinity), such as the decrease in sea ice temperature over two

days when snow cover was removed from an area at Station Nord, could result in increased ikaite concentrations, even over short time scales. This is also consistent with observations at Young Sound, Greenland, where ikaite crystals were observed in new sea ice with one hour (Rysgaard et al., 2013). It is likely that the $CO_2$ produced by ikaite precipitation near the upper sea ice surface, such as in the upper layers of the snow cleared area at Station Nord, is released to the atmosphere, but the overall impact of this is likely minor (Geilfus et al., 2013a). Increased

ikaite concentrations over a short period at Station Nord indicate that if ikaite continuously precipitates and dissolves throughout the sea ice season, especially near the upper ice surface where changes in temperature are more likely due to variable air temperatures and snow cover, the sea ice driven carbon pump will be more significant throughout the winter.

Snow cover on sea ice can impact sea ice temperature and will likely also influence ikaite precipitation. Ikaite

concentrations were lower in first–year sea ice at Station Nord than at Cambridge Bay since the sea ice was insulated from the atmosphere at Station Nord due to the thicker snow cover (~1 m), resulting in warmer sea ice temperatures. At Cambridge Bay, snow cover had less of an influence on ikaite concentration than at Station Nord.





Temperatures in the upper sea ice layers at Cambridge Bay were similar to air temperatures, suggesting that the insulating effect of the snow cover was limited. As a result of the decreased length of the sea ice season, snow depth on Arctic sea ice is decreasing (Hezel et al., 2012) so sea ice temperatures could also fluctuate with air temperature if the insulating effect of snow is less pronounced, such as the decreasing ice temperature in the snow cleared area at Station Nord. In this area, ikaite concentrations increased, indicating that ikaite could continuously precipitate and dissolve with temperature fluctuations throughout the winter, which could potentially enhance $CO_2$ fluxes during the winter months (Rysgaard et al., 2013). Colder sea ice resulting from a thinner snow cover can also decrease sea ice permeability, increasing the likelihood that ikaite remains trapped in the sea ice matrix and will contribute to the sea ice carbon pump. In contrast, a thick snow cover may impede air–sea $CO_2$ exchange (Brown et al., 2015), resulting in a release of $CO_2$ to the atmosphere when the snow melts.

Ikaite has been observed in snow but the distribution of ikaite in snow is unknown, although it is likely that crystals are transported into the snow cover from the sea ice (Fischer et al., 2013). It is necessary to compare sea ice with similar salinities and variable snow cover before the overall impact of snow cover on ikaite precipitation, the sea ice carbon pump, and air–sea $CO_2$ fluxes can be fully understood.

## 5 Conclusions

Based on the results from the Station Nord and Cambridge Bay field campaigns, DIC analysis of filtered crystals is an effective and efficient method of determining ikaite concentration in sea ice that agrees with existing techniques. The DIC method is a simple technique that yields more accurate and more precise ikaite concentrations than image analysis in all sea ice types and we recommend it be used in future ikaite quantification measurements in sea ice.

Ikaite concentrations in sea ice are affected by temperature, salinity, TA, DIC, and the TA:DIC ratio. Based on the data collected from SERF, Station Nord, and Cambridge Bay, we can make several conclusions regarding the influence of these environmental parameters on ikaite concentrations:

1. Ikaite will precipitate most readily in cold sea ice with high bulk salinities and high TA:DIC ratios. The highest ikaite concentrations were observed at SERF, where ikaite concentrations in the upper layers of sea ice approached 1000 μmol kg$^{-1}$, sea ice temperatures were below –3.9ºC, bulk salinities were above 4.9, TA and DIC were above 376.2 and 316.9 μmol kg$^{-1}$, respectively, and the TA:DIC ratio approached 1.7. Conversely, ikaite concentrations in first– and multi–year sea ice at Station Nord, where sea ice temperatures were warmer and bulk salinities, TA and DIC concentrations, and the TA:DIC ratio were lower, were all below 100 μmol kg$^{-1}$.

2. Ikaite concentrations at Station Nord were lower than at Cambridge Bay and at other locations in the Arctic. The low salinity seawater at Station Nord in conjunction with the low sea ice bulk salinity, TA, and DIC likely prevent high ikaite concentrations.



3. Ikaite and sea ice algae were observed together at Cambridge Bay for, to our knowledge, the first time. The presence of sea ice algae may enhance the conditions that promote ikaite precipitation since algae consume carbon during photosynthesis, increasing pH.

Replacement of multi-year ice with seasonal sea ice in the Arctic will result in higher salinities and TA and DIC concentrations. Based on the observations made during this study, it is likely that this will enhance ikaite precipitation, particularly during the growth of new sea ice. The presence of ikaite and sea ice algae together indicates that increased sea ice algae production resulting from thinner sea ice cover may increase ikaite concentrations in the bottom ice layers and further enhance the role of ikaite in the sea ice carbon pump.

Elevated ikaite concentrations will result in increased $CO_2$ production. If the $CO_2$ is removed from the sea ice and ikaite remains trapped within the sea ice matrix, then TA stored in ikaite will increase, which will enhance the buffering capacity of the sea ice and meltwater when ikaite dissolves during ice melt and TA is released. This will result in a more efficient sea ice carbon pump and increased $CO_2$ uptake by the surface waters. Using the new technique to quantify ikaite will lead to increased understanding of its spatial and temporal dynamics as well as its role in carbon fluxes in ice–covered seas.

**6 Acknowledgements**

We gratefully acknowledge the contributions of the Canada Excellence Research Chair (CERC) and Canada Research Chair (CRC) programs. Support was also provided by the Natural Sciences and Engineering Research Council (NSERC), the Canada Foundation for Innovation, and the University of Manitoba. This work is a contribution to the ArcticNet Networks of Centres of Excellence and the Arctic Science Partnership (ASP; asp-net.org).



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



**Tables**

Table 1: Geographical locations and types of ice collected during the SERF 2013, Station Nord 2015, and Cambridge Bay 2016 field campaigns.

| Ice type | Sampling site | | |
|---|---|---|---|
| | *SERF 2013* | *Station Nord 2015* | *Cambridge Bay 2016* |
| First–year sea ice | | x | x |
| Multi–year sea ice | | x | |
| Artificial sea ice | x | | |
| **Position** | 49ºN 97ºW | 81ºN 17ºW | 69ºN 104ºW |
| **Sampling days** | 23 January, 2013 | 10 April–3 May, 2015 | 27 April–24 May, 2016 |



Table 2: Average sea ice thicknesses and ikaite concentrations in first– and multi–year sea ice determined using image analysis and DIC analysis.

| Sampling site | Average ice thickness (cm) | Ikaite concentration ($\mu$mol kg$^{-1}$), image analysis method | Ikaite concentration ($\mu$mol kg$^{-1}$), DIC method |
|---|---|---|---|
| SERF | 20 | 334.9±323.1 | – |
| Station Nord, FYI | 122.0 | 7.0±1.2 | 5.1±0.8 |
| Station Nord, MYI | 165.5* | 0.2±0.1 | 3.1±0.3 |
| Cambridge Bay | 173.8 | 73.4±26.2 | 44.6±3.0 |

*Due to the limitations of the coring equipment, it was not possible to collect full multi–year sea ice cores.

Errors represent the standard error of the mean.



**Figures**

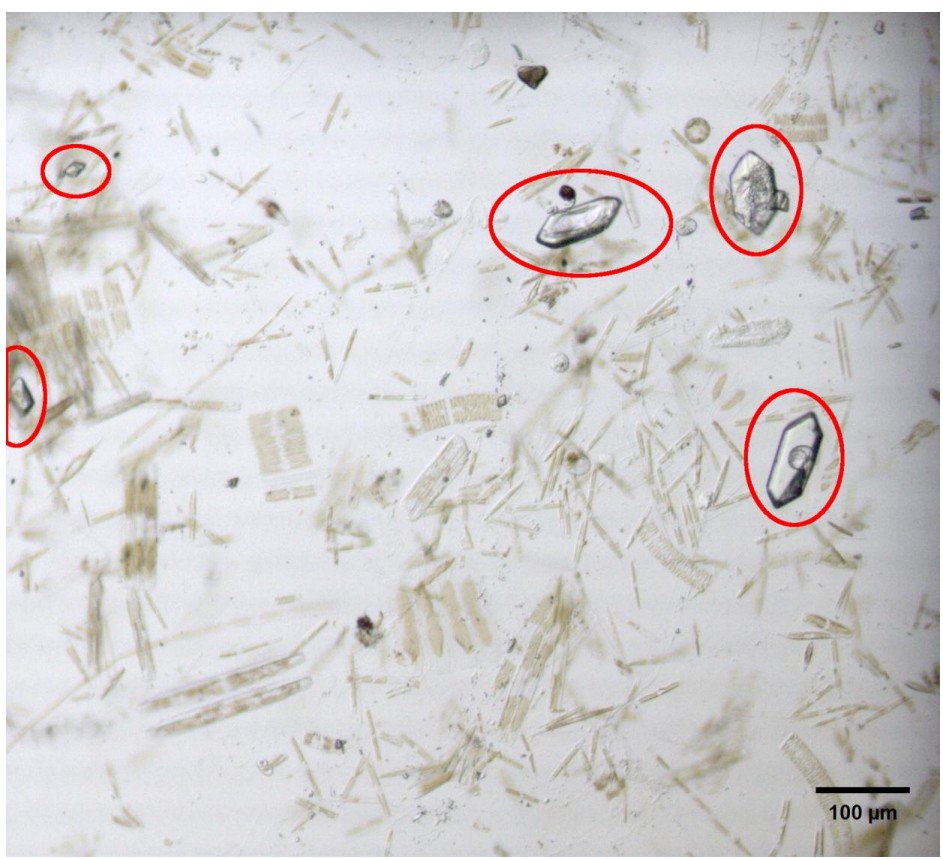

Figure 1: Image of ikaite crystals (circled in red) and sea ice algae observed together near the bottom of sea ice core
CB–2016–06, Cambridge Bay, Nunavut, May 2016.





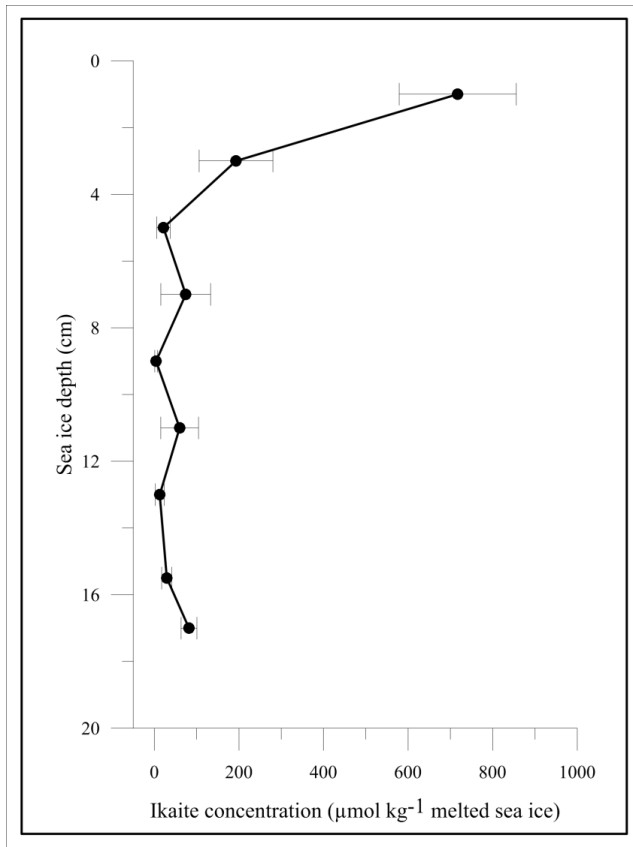

Figure 2: Average ikaite concentrations in artificial sea ice calculated using the image analysis method, SERF, January 2013. All error bars represent the standard error of the mean.




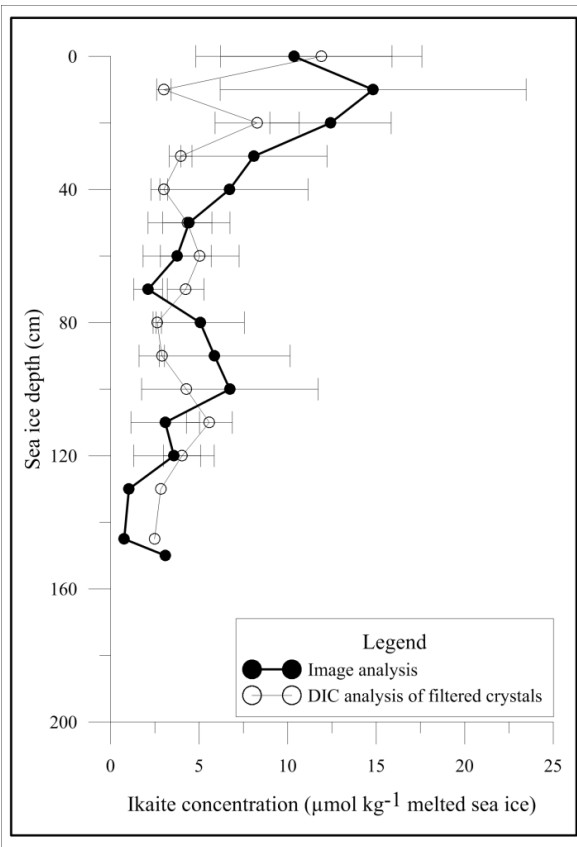

Figure 3: Average ikaite concentrations in first–year sea ice calculated using image analysis and DIC analysis of
filtered crystals, Station Nord, Greenland, April 2015. All error bars represent the standard error of the mean.





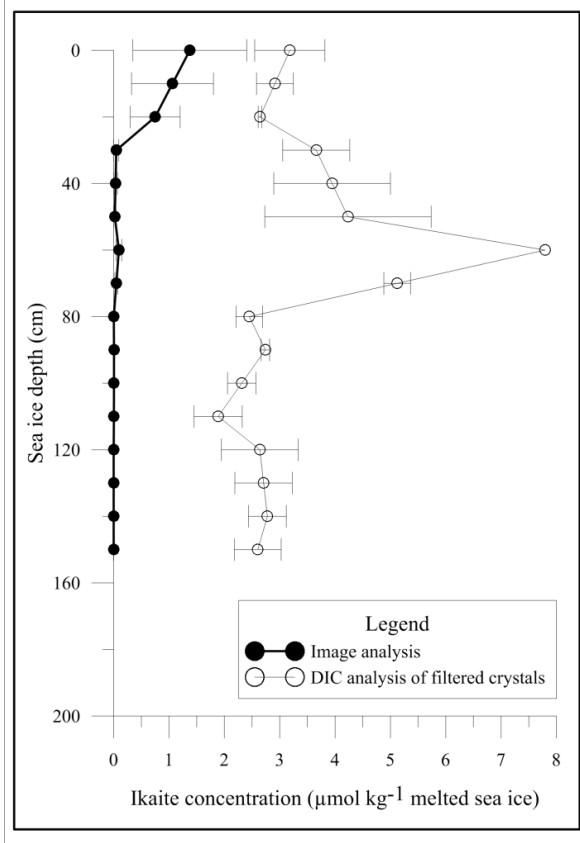

Figure 4: Average ikaite concentrations in multi–year sea ice calculated using image analysis and DIC analysis of
filtered crystals, Station Nord, Greenland, April 2015. All error bars represent the standard error of the mean.



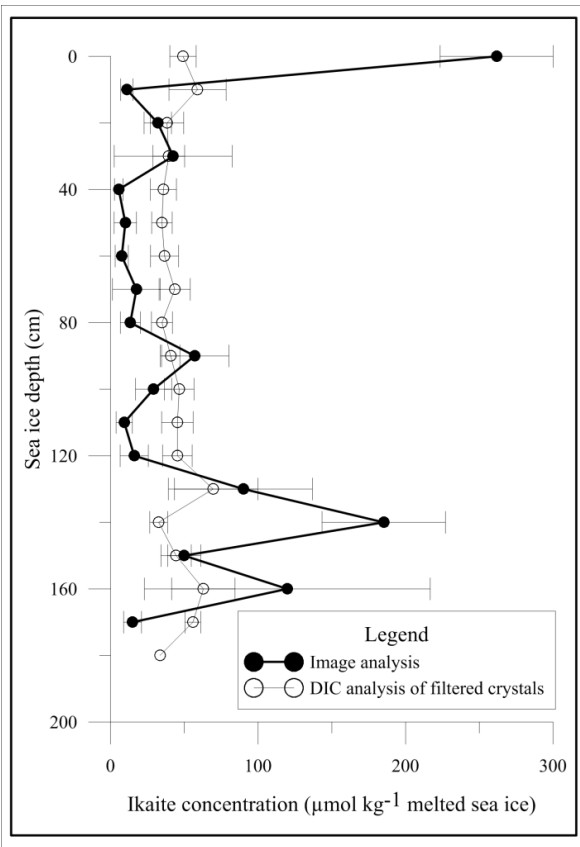

Figure 5: Average ikaite concentrations in first–year sea ice calculated using image analysis and DIC analysis of filtered crystals, Cambridge Bay, Nunavut, May 2016. Error bars for concentrations determined using image analysis in Fig. 5 represent the standard error of the mean except at 0 cm (0.25 standard error) and 140 cm (0.25 standard error). All error bars for concentrations determined using DIC analysis of filtered crystals in Fig. 5 represent the standard error of the mean.





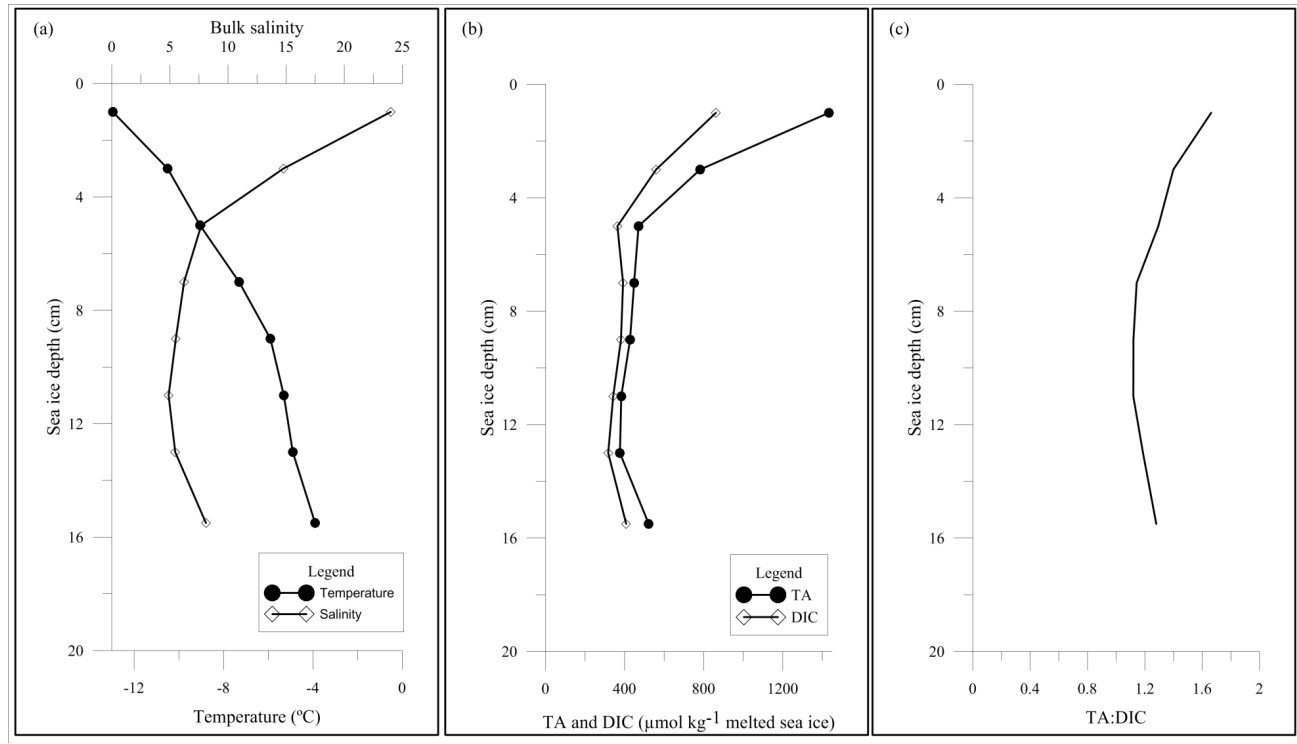

Figure 6: Vertical profiles of (a) temperature and bulk salinity, (b) TA and DIC concentrations, and (c) the TA:DIC ratio in artificial sea ice, SERF, January 2013.




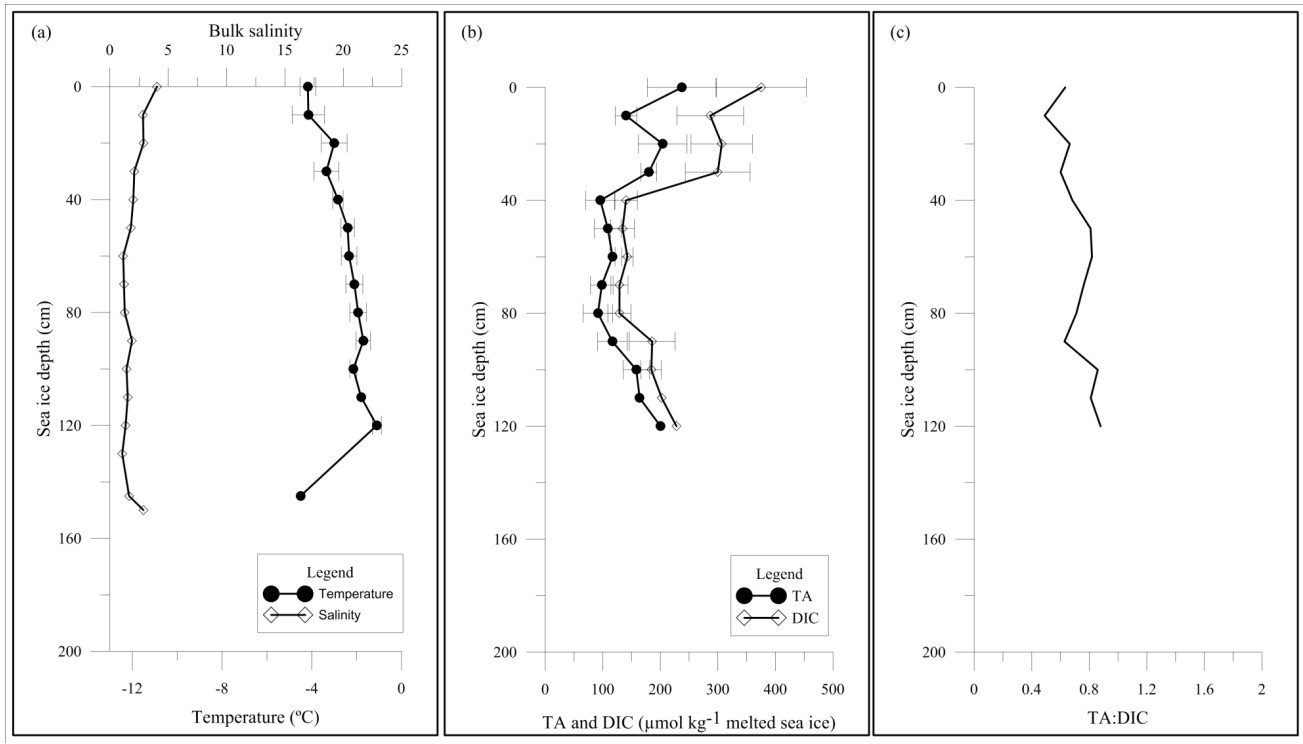

Figure 7: Average vertical profiles of (a) temperature and bulk salinity, (b) TA and DIC concentrations, and (c) TA:DIC in first–year sea ice, Station Nord, Greenland, April 2015. All error bars represent the standard error of the mean.



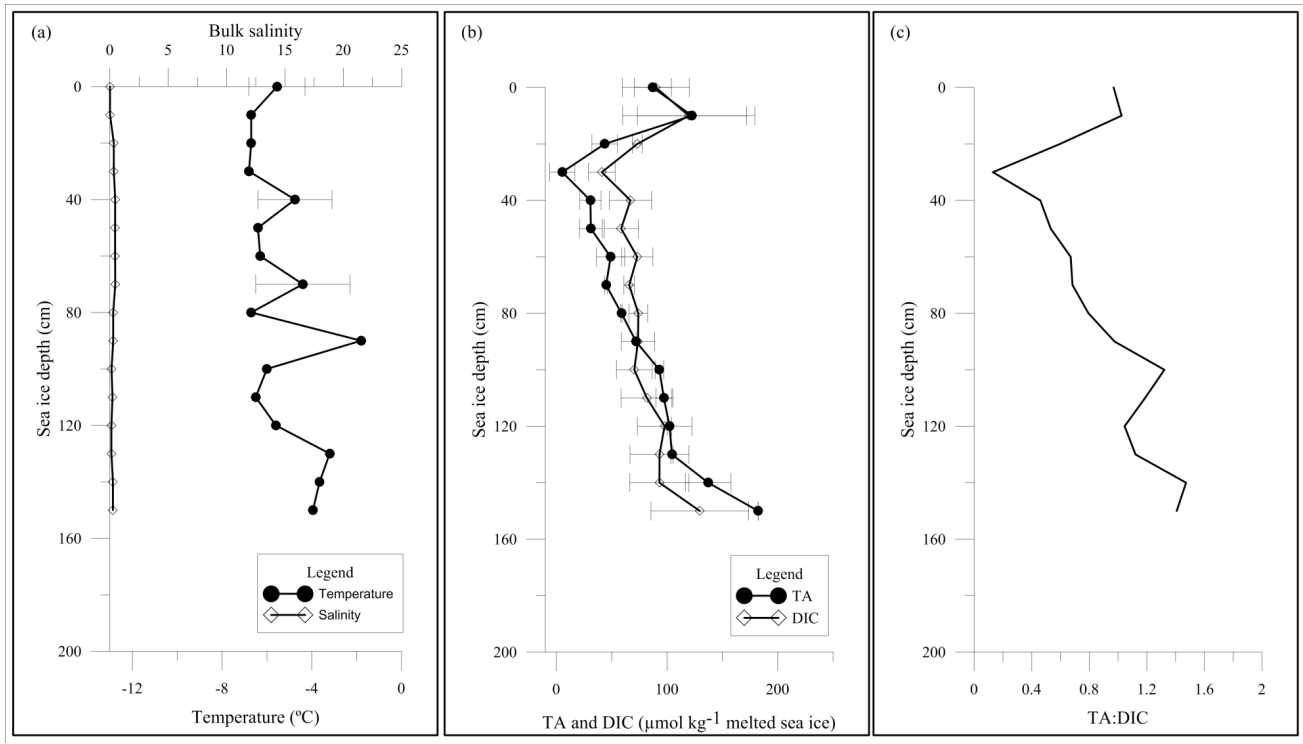

Figure 8: Average vertical profiles of (a) temperature and bulk salinity, (b) TA and DIC concentrations, and (c) TA:DIC in multi–year sea ice, Station Nord, Greenland, April 2015. All error bars represent the standard error of the mean.





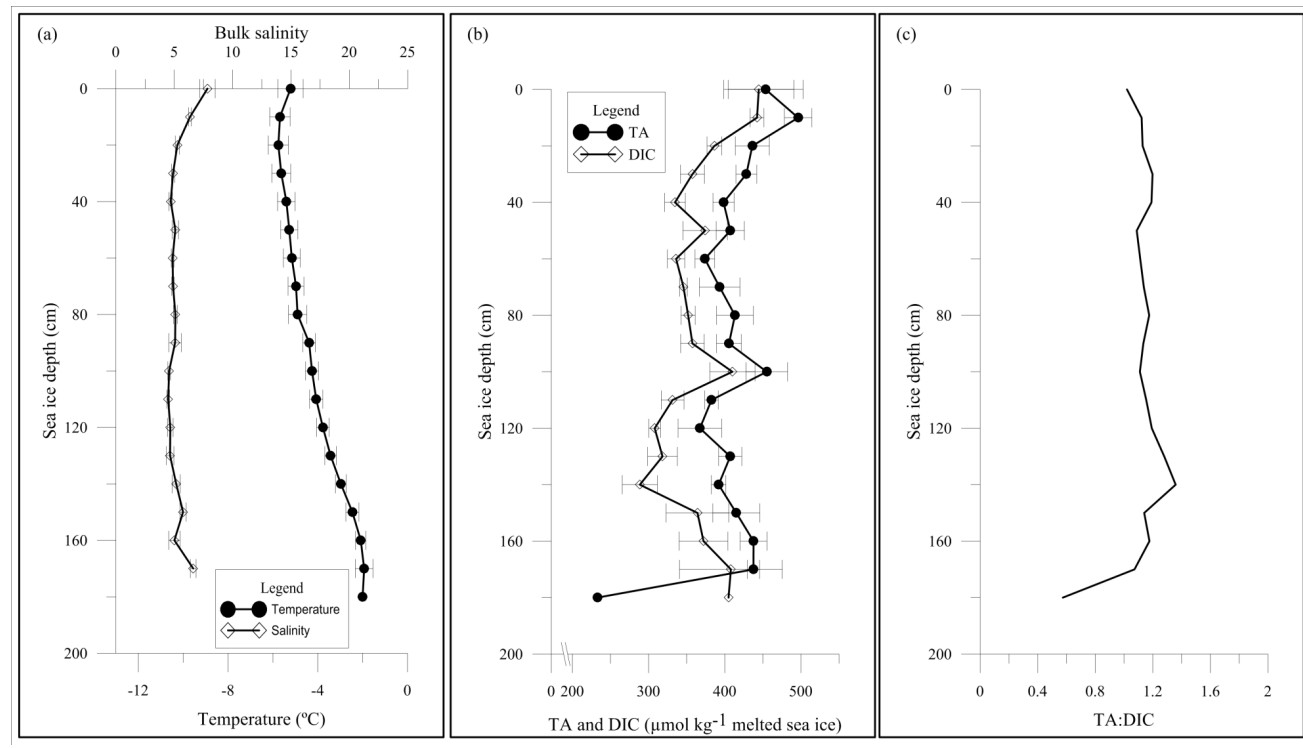

Figure 9: Average vertical profiles of (a) temperature and bulk salinity, (b) TA and DIC concentrations, and (c) TA:DIC in first–year sea ice, Cambridge Bay, Nunavut, May 2016. All error bars represent the standard error of the mean.



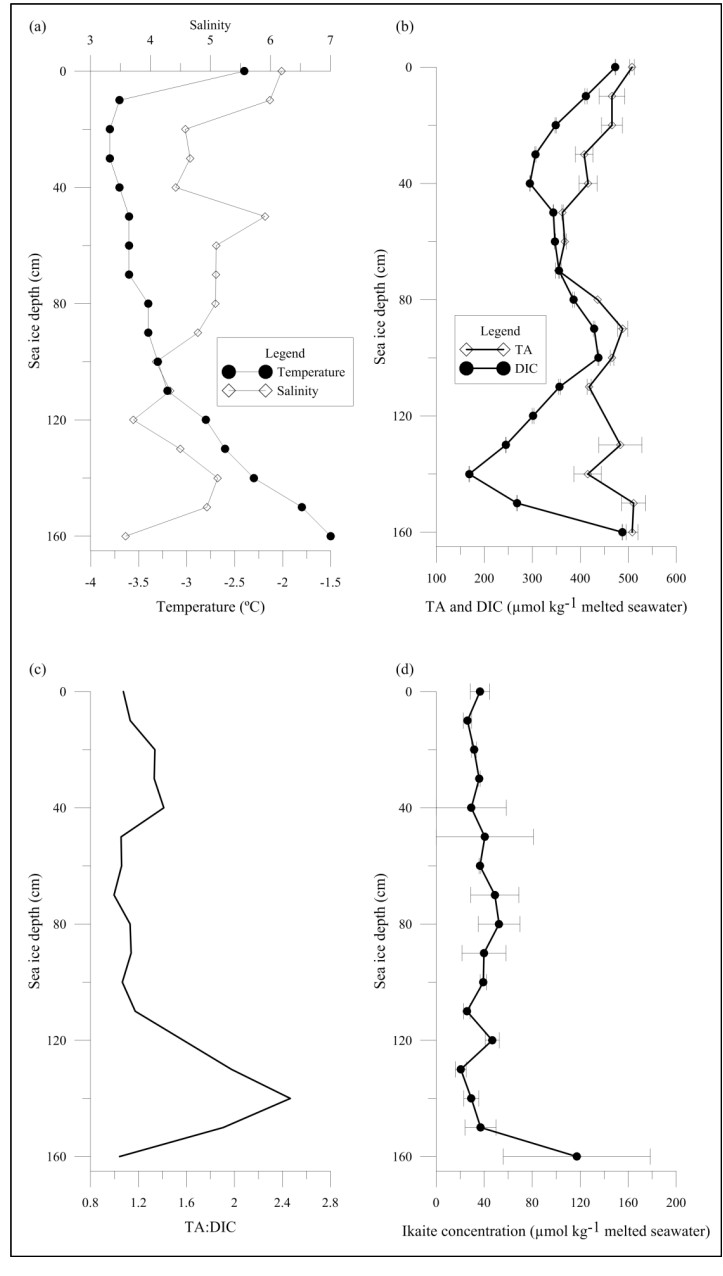

Figure 10: Vertical profile for sea ice core CB–2016–06: (a) temperature and bulk salinity; (b) TA and DIC concentrations; (c) the TA:DIC ratio; and (d) ikaite concentration. The decrease in DIC and related increase in the TA:DIC ratio is associated with the presence of sea ice algae near the bottom of the core. Error bars represent the standard error of the mean.