# Peer review of "Quantification of calcium carbonate (ikaite) in first- and multiyear sea ice"

_The Cryosphere, 2017_

## Referee Comment (RC1) · Anonymous Referee #1 · 26 Dec 2017

General comments

Heather Kyle et al. quantified ikaite in sea ice by new method. However, this method contains the possibility to do over estimation the ikaite within sea ice because in fact, sea ice contains the many particles (e.g. CaCO3 contained dust, sediment, phytoplankton etc). Also, difference of ikaite amount between methods changed dramatically for each type of sea ice. Therefore, we cannot quantify the accurate amount of ikaite in sea ice by using new method, and this method cannot apply for various types of sea ice in the polar oceans. In addition, the explanation of TA:DIC ratio in melted ice water was not clear although it is important part in this paper to indicate precipitation of ikaite within sea ice.

Specific comments

line 20: High presser within sea ice?

line 41: Only Arctic?

lines 45-46: Why DIC etc decrease when dense brine sink?

line 45, TA is not concentration.

line 66: Suddenly you used pCO2, but before you used CO2 concentration (line 46). What differences between them?

line 81: All particles could trapped in filter, not only ikaite but also CaCO3 contained particles (dust, sediments, algae etc). Therefore, I strongly disagree about this method to quantify the ikaite. If you will quantify PIC (particulate inorganic carbon), I agree.

lines 95-96: You should indicate the chemical properties of artificial seawater used at SERF. The amount of ikaite was clearly high as compared to other natural ice. The DIC:TA of ice depend on the seawater properties. Therefore, also please indicate TA and DIC values. In addition, there was no comparison between image method and filtered method (Only used image method). Therefore, this data does not need in this paper.

line 105: You kept samples in freezer. Therefore, we cannot believe that it is real ikaite amount. As you mentioned in discussion (lines 307-309, 376-377), ikaite precipitates at short time scale.

lines 106-107: What kind of gas tight bags? If CO2 coming from outside, ikaite amount would be changed as you mentioned. Therefore, you have to indicate results of gas tight test.

line 138-139: why did you take water sample before filtered? If this water contains ikaite crystals, results will be changed.

line 140: how to do filtration? If filter was vacuumed, DIC will change, and it is not recommended (Miller et al., 2015).

line 170: how statistically agree between each method? You can make 1:1 relationship figures (e.g. image vs filtered) to help leader.

lines 207-208: Under-ice water DIC and TA is need to think what happened in sea ice during sea ice formation.

lines 223-232: I can not understand this comparison. Did you compare the same piece (section) of sea ice sample? You checked for one section by image then filtered same water for filtered method? OR Different piece for each method?

line 223-232: Did you calculate standard error for same samples? Or many ice sections for each method although you indicated section size for image analysis from line 256?

lines 233-246: Again how about the possibility of CaCO3 contained particles?

lines 320-327: When you will make figure about TA vs DIC, can you see the CaCO3 formation from seawater's DIC and TA? Based on this figure, you can also calculate the ikaite amount within sea ice.

line 320-327: You measured TA and DIC for melted-ice water without containing ikaite crystals (Line 138-140."As soon as melting was complete, four 12 ml Exetainers (Labco Limited, High Wycombe, UK) were filled with meltwater for DIC and total alkalinity (TA) analysis"). If so, TA:DIC should low because ikaite precipitation removed TA than DIC from melted water (if ikaite crystals remained, TA remained in crystals, meaning that water TA was low). If you measured TA and DIC after all ikaite was dissolved, I understand that TA: DIC increased with respect to before (e.g. seawater values) when ikaite is precipitated.

line 334-347: TA:DIC ratio changed by many processes (e.g. ikaite, biology, gas exchange). Therefore, drawing DIC vs TA provides detailed discussion.

Figures 2-10: Black line at the outside of figure do not need.

Figures 3-5, Error bar is similar length for each method. But you mentioned that standard error for image method was higher than that for filtered method.

Figure 4: Does this difference indicate PIC?

Figure 8: Why sea ice temperature changed dramatically?

Figures 6-9: why no points for TA:DIC profile? And no error bar?

Figure 10d. why only one method? If only one method, we do not need this figure and not important for this method paper.

---

## Referee Comment (RC2) · Anonymous Referee #2 · 26 Feb 2018

Ikaite is a peculiar mineral intrinsically intriguing to many people with diverse background. So I understand the interest in studying this mineral in sea ice and I enjoyed some of the early ikaite papers about sea ice. After reading through a series of papers by the senior author cited in the introduction (ln55-65), I'm still not convinced at all that ikaite abundance in sea ice has the impact anywhere close to what they are claiming it to be. Overhyped implications of ikaite on carbon cycle do not seem to be a wise choice of motivation for this type of study. This is almost like arguing that it is critical to study ikaite in frozen shrimp due to its global health effect. The data may have some value in them, but I feel the senior author should seriously brainstorm for the real novelty of such investigations. I won't further comment on the technical details of the paper, because I don't want the graduate student to feel overly disheartened.

---

## Author Comment (AC1) · 27 Feb 2018

**Author's response to Anonymous Referee #1:**

We thank the reviewer for their comments. Our responses are below in red.

**General comments:**

Heather Kyle et al. quantified ikaite in sea ice by new method. However, this method contains the possibility to do over estimation the ikaite within sea ice because in fact, sea ice contains the many particles (e.g. CaCO3 contained dust, sediment, phytoplankton etc). Also, difference of ikaite amount between methods changed dramatically for each type of sea ice. Therefore, we cannot quantify the accurate amount of ikaite in sea ice by using new method, and this method cannot apply for various types of sea ice in the polar oceans. In addition, the explanation of TA:DIC ratio in melted ice water was not clear although it is important part in this paper to indicate precipitation of ikaite within sea ice.

We thank you for your comments. However, we disagree that the DIC method will overestimate ikaite concentrations. Previous studies have suggested that the presence of $CaCO_3$ in ice could originate from calcareous components of rock flour and wind-blown dust (Killawee et al., 1998) or incorporated by sediment impurities (Eicken, 2004). However, atmospheric transport and transformations to the Arctic were quantified in the Danish AMAP program, and although $CaCO_3$ was not specifically included in this study, $Ca^{2+}$ only occurred in very low concentrations and originated partly from sea spray and soil (Heidam et al. , 2004). In addition, the coring sites in the present study did not contain any visible impurities when melted in our bags and our filters. Furthermore, a comprehensive study of the sediments in the Russian Arctic (Kosheleva, 2002) showed low concentrations of CaCO3 (<2.4%). Also, surface sediments of the Greenland Sea/Nordic Seas are extremely poor in carbonates (Huber et al., 2000; Hebbeln et al., 1998; Rysgaard and Glud, 2007). We have incorporated this into the manuscript to make this more clear.

Finally, samples that contained large amounts of sea ice algae did not yield significantly different results than those that did not, leading us to conclude that the presence of particulate carbon did not significantly impact measured ikaite concentrations. Since we did not measure seawater TA and DIC, we are not able to quantify the different factors that influence the TA:DIC ratio such as ikaite precipitation, gas exchange, and biology. This is discussed more in the responses to your specific comments.

**Specific comments:**

line 20: High presser within sea ice?

Ikaite does not form exclusively within sea ice. In sea ice, it forms only at low temperature rather than at high pressure. We rephrased this part of the manuscript to clarify that we are focussing only on ikaite precipitation in sea ice.

line 41: Only Arctic?

The study focussed on Arctic sea ice. Ikaite concentration may increase in Antarctic sea ice as well, but our study cannot confirm this, so we did not change this in the manuscript.

lines 45-46: Why DIC etc decrease when dense brine sink?

DIC is removed from the sea ice along with the brine that is heavy and sink into the underlying water column, so DIC concentrations of the sea ice and surface waters decrease (Rysgaard et al., 2007). We added a sentence to the manuscript to make this more clear.

line 45, TA is not concentration.

We have changed this here and elsewhere in the manuscript so TA is not referred to as a concentration.

line 66: Suddenly you used pCO2, but before you used CO2 concentration (line 46). What differences between them?

The *'p'* stands for partial pressure. We corrected line 46 to refer to $pCO_2$ as well.

line 81: All particles could trapped in filter, not only ikaite but also CaCO3 contained particles (dust, sediments, algae etc). Therefore, I strongly disagree about this method to quantify the ikaite. If you will quantify PIC (particulate inorganic carbon), I agree.

With the exception of sea ice sections with visible sea ice algae, very few particles were observed on the filters. In addition, sediments and wind-blown dust throughout the Arctic are poor in calcium carbonates, as discussed in the response to the general comments above.

lines 95-96: You should indicate the chemical properties of artificial seawater used at SERF. The amount of ikaite was clearly high as compared to other natural ice. The DIC:TA of ice depend on the seawater properties. Therefore, also please indicate TA and DIC values. In addition, there was no comparison between image method and filtered method (Only used image method). Therefore, this data does not need in this paper.

We added a reference to Hare et al. (2013) to the description of the SERF pool that includes a description of the chemical properties of the seawater. We plan to leave the SERF data in the paper since they support discussion sections 4.2 and 4.3, which focus on the relationship between ikaite and other environmental parameters, the influence of ikaite in ice covered seas, and how ikaite precipitation will change with changing sea ice conditions.

line 105: You kept samples in freezer. Therefore, we cannot believe that it is real ikaite amount. As you mentioned in discussion (lines 307-309, 376-377), ikaite precipitates at short time scale.

We agree. However, it is not possible to complete image analysis without storing at least some of the samples in the freezer since it is so time consuming. The samples used for DIC analysis were not stored in the freezer. We reworded lines 105-107 slightly to make this clear. The difference in storage times may partly account for discrepancies between the ikaite concentrations determined using each method, particularly when image analysis yields significantly higher concentration variation than whole section DIC analysis. This has been incorporated into the discussion.

lines 106-107: What kind of gas tight bags? If CO2 coming from outside, ikaite amount would be changed as you mentioned. Therefore, you have to indicate results of gas tight test.

The gas tight bags used were nylon/poly bags from Cabela's and sealed using the method from Hu et al. (2017). We added this to the methods section.

line 138-139: why did you take water sample before filtered? If this water contains ikaite crystals, results will be changed.

Water samples were taken before filtration because we were looking at bulk sea ice conditions, which include ikaite crystals.

line 140: how to do filtration? If filter was vacuumed, DIC will change, and it is not recommended (Miller et al., 2015).

Filtrations were completed using a vacuum hand pump. Testing by Hu et al. (2017) determined that using a vacuum pump to evacuate air during DIC sample preparation yielded no statistically significant difference from samples where a vacuum was not applied. We therefore disagree that our method will change the DIC concentration.

line 170: how statistically agree between each method? You can make 1:1 relationship figures (e.g. image vs filtered) to help leader.

Statistical analysis was completed using a paired t-test (p-value = 0.05), which is indicated on line 229-230. More details regarding the agreement between methods are given in discussion section 4.1.

lines 207-208: Under-ice water DIC and TA is need to think what happened in sea ice during sea ice formation.

We are not sure what you mean here. We are not speculating on what happened during sea ice formation and are instead focussing on the current sea ice conditions.

lines 223-232: I can not understand this comparison. Did you compare the same piece (section) of sea ice sample? You checked for one section by image then filtered same water for filtered method? OR Different piece for each method?

One core was used for the DIC method. The second core from each sampling site was used for image analysis then melted and used for duplicate samples for the DIC method. This is outlined in the methods section (lines 130-135).

line 223-232: Did you calculate standard error for same samples? Or many ice sections for each method although you indicated section size for image analysis from line 256?

We are not sure what you mean here. We calculated the standard error of the mean as shown in the different figures.

lines 233-246: Again how about the possibility of CaCO3 contained particles?

See previous comment above.

lines 320-327: When you will make figure about TA vs DIC, can you see the CaCO3 formation from seawater's DIC and TA? Based on this figure, you can also calculate the ikaite amount within sea ice.

We plotted TA vs. DIC but chose not to include the figure in the manuscript since we did not record the seawater TA and DIC and we did not focus on other factors that will influence sea ice TA and DIC. The TA:DIC ratio of seawater is typically around 1.0 and the TA:DIC ratios above 1 in the sea ice indicate that more ikaite is present here.

line 320-327: You measured TA and DIC for melted-ice water without containing ikaite crystals (Line 138-140."As soon as melting was complete, four 12 ml Exetainers (Labco Limited, High Wycombe, UK) were filled with meltwater for DIC and total alkalinity (TA) analysis"). If so, TA:DIC should low because ikaite precipitation removed TA than DIC from melted water (if ikaite crystals remained, TA remained in crystals, meaning that water TA was low). If you measured TA and DIC after all ikaite was dissolved, I understand that TA: DIC increased with respect to before (e.g. seawater values) when ikaite is precipitated.

It is assumed that ikaite crystals did not dissolve during sea ice melt. Sampling of the meltwater for TA and DIC analysis was done before filtration, so the samples still contained ikaite. It is assumed that since the samples warm to above 4ºC prior to analysis, ikaite will dissolve and the TA and DIC stored within the crystals will dissolve and will be measured during analysis.

line 334-347: TA:DIC ratio changed by many processes (e.g. ikaite, biology, gas exchange). Therefore, drawing DIC vs TA provides detailed discussion.

See comment above.

Figures 2-10: Black line at the outside of figure do not need.

We agree that these lines are unnecessary so they will be removed.

Figures 3-5, Error bar is similar length for each method. But you mentioned that standard error for image method was higher than that for filtered method.

The error bars for the image analysis are indeed larger than those for the DIC method, particularly in first year ice (Figs. 3 and 5). In order to show the difference in standard error more clearly, we will represent the error bars for each technique in different colours.

Figure 4: Does this difference indicate PIC?

We don't think so based on the arguments above. However, it is likely that in this case image analysis is underestimating ikaite concentration since the crystals were too small to observe using our imaging technique. This is discussed in the body of the text (lines 233-246).

Figure 8: Why sea ice temperature changed dramatically?

Figure 8 represents multi-year sea ice. Due to the large difference between air and sea ice temperatures at Station Nord, it was difficult to obtain full temperature profiles, which we made clear in the results section. It is not expected that multi-year sea ice will have the same temperature profile as first-year ice, so the temperature variation in this figure is not significant.

Figures 6-9: why no points for TA:DIC profile? And no error bar?

The points and error bars were not placed in the TA:DIC profiles and have been added.

Figure 10d. why only one method? If only one method, we do not need this figure and not important for this method paper.

The purpose of this figure is to highlight the increase in ikaite concentration associated with the presence of sea ice algae in the bottom of the ice at Cambridge Bay (lines 348-357) so we disagree that the figure should not be included. We only showed one method in the figure to avoid confusion.

---

## Author Comment (AC2) · 1 Mar 2018

**Response to Anonymous Referee #2**

We find this review not to be very constructive. However, we will try to answer it to the best of our ability.

The Referee states that he/she is not convinced that ikaite abundance in sea ice has the impact anywhere close to what former senior author claims it to be (ln 55-65). First, this paragraph is included to explain how the presence of ikaite (formation and dissolution) can affect the carbonate system. This will naturally have an effect on the exchange of $CO_2$ between the ocean and the atmosphere and the pH of surface waters. This background information is needed to understand the dynamics of ikaite in sea ice and how it can modify the exchange between the atmosphere and ocean. Thus, it is important to keep this section in the manuscript. Second, this manuscript is not dealing with the global implications of ikaite, but it presents a new and novel technique that will allow for more measurements of ikaite from different regions. So far, there have been few measurements of ikaite from few geographical regions. Third, we find that sea ice algal communities seem to stimulate ikaite formation, something never before reported. Finally, we do not agree that we 'overhype' the implications of ikaite. Throughout the manuscript the wording we use is "may play a significant role" (e.g., ln 61). A few years ago we did not know ikaite existed in sea ice. Since then we have been investigating the details of ikaite formation and dissolution in sea ice, as well as the potential effects of ikaite on the atmosphere-ocean exchange of $CO_2$ at local, regional and global scales. Locally and regionally we have found that ikaite can explain a large part of the atmosphere-ocean $CO_2$ flux (e.g. Rysgaard et al., 2009), which matches regional outputs from global simulations (Grimm et al., 2016). Our recent model results show that the flux on a global scale is minor as the $CO_2$ taken up by the Arctic seas is released again to the atmosphere further south. As the uptake of this ikaite mediated $CO_2$ flux in the global model is fully linked to brine formation and the spatial resolution of the global models (both resulting in a poorly constrained brine dynamic in these models) there is still room for new discoveries to be made. The new method we report here will contribute to such discoveries and to quantitatively examine their significance.

The final comment by the referee that he/she will not further comment on the technical details of the paper because he/she does not want the graduate student to feel overly disheartened is noted. It would have been more appropriate and perhaps productive to contact the senior author directly by email or phone and discuss his/her frustrations rather than providing an anonymous comment in a public forum. Perhaps we could have had a scientific discussion on the matter that could have benefitted the scientific community. Disagreement is often a way to learn new things☺.

**References:**

Grimm, R., Notz, D., Glud, R.N., Rysgaard, S., and Six, K.D.: Assessment of the sea–ice carbon pump: Insights from a three–dimensional ocean–sea–ice biogeochemical model (MPIOM/HAMOCC), Elementa: Science of the Anthropocene, 4, doi:10.12952/journal.elementa.000136, 2016.

Rysgaard, S., Bendtsen, J., Pedersen, L.T., Ramløv, H., and Glud, R.N.: Increased CO2 uptake due to sea ice growth and decay in the Nordic Seas, J. Geophys. Res., 114, C09011, doi:10.1029/2008JC005088. 2009.

---

## Editor Comment (EC1) · T. Maksym (Editor) · 2 Mar 2018

Based on the critical reviews and the author response, I will provide here some additional guidance for the authors as they prepare a revised manuscript. I see three main points raised by the reviewers that should be addressed if the paper is to be accepted:

(1) The question of other sources of DIC. Given the large discrepancy between methods for some samples, this should be thoroughly addressed. The authors do provide several references that may be sufficient to rebut this claim. I do urge the authors to provide a quantitative estimate of the potential contribution to DIC from non-Ikaite sources.

(2) I agree with reviewer #1 that there is a fair amount of discrepancy between the

methods for different cores. This is a serious weakness if the paper is claiming the new method matches the old. The authors do not really address this comment in their response. The authors do provide possible explanations for these discrepancies in the original text, but the reviewer has a point that this then makes validation of the new method difficult. I suggest that the authors either spend more time justifying this purported match (e.g. the 1:1 plot suggestion of the reviewer), or, if they are claiming the new method is superior for some cases (e.g. issue (1) above is not significant, while issues with image analysis techniques are) then to make this argument instead.

(3) Reviewer #2 argues against the importance of Ikaite in the carbon cycle, and hence the potential impact of this study. The authors provide a strong response and clarify the motivation for the study (which should be made clear in the revised manuscript). The contention is perhaps over the interpretation of "may be significant", which is a vague and not very informative term. I urge the authors to consider how they might better convey the potential importance of Ikaite in the introduction in more quantifiable terms if possible.

When the authors submit a revised manuscript, it will be sent out for additional review.